# Revisiting Periodontal Disease in Dogs: How to Manage This New Old Problem?

**DOI:** 10.3390/antibiotics11121729

**Published:** 2022-12-01

**Authors:** Eva Cunha, Luís Tavares, Manuela Oliveira

**Affiliations:** 1CIISA—Center for Interdisciplinary Research in Animal Health, Faculty of Veterinary Medicine, University of Lisbon, Avenida da Universidade Técnica, 1300-477 Lisboa, Portugal; 2Associate Laboratory for Animal and Veterinary Sciences (AL4AnimalS), 1300-477 Lisboa, Portugal

**Keywords:** periodontal disease, dogs, dental plaque

## Abstract

Periodontal disease (PD) is one of the most prevalent oral inflammatory diseases in dogs. PD onset begins with the formation of a polymicrobial biofilm (dental plaque) on the surface of the teeth, followed by a local host inflammatory response. To manage this disease, several procedures focusing on the prevention and control of dental plaque establishment, as well as on the prevention of local and systemic PD-related consequences, are essential. The removal of dental plaque and the inhibition of its formation can be achieved by a combination of dental hygiene homecare procedures including tooth brushing, the application of different oral products and the use of specific diet and chew toys, and regular professional periodontal procedures. Additionally, in some cases, periodontal surgery may be required to reduce PD progression. Associated with these measures, host modulation therapy, antimicrobial therapy, and other innovative therapeutic options may be useful in PD management. Moreover, PD high prevalence and its relation with potential local and systemic consequences reinforce the need for investment in the development of new preventive measures, treatments, and oral procedures to improve the control of this disease in dogs. Knowledge on the specific guidelines and diversity of the available products and procedures are fundamental to apply the most adequate treatment to each dog with PD.

## 1. Introduction

Periodontal disease (PD) is one of the most common inflammatory diseases in dogs. It affects the periodontium, leading to gingivitis and/or several degrees of periodontitis [1,2,3]. PD is initiated by the formation of a pellicle, composed of a saliva-derived layer of glycoproteins that coats the surface of the tooth, followed by bacterial adherence to this layer, resulting in a polymicrobial biofilm, also known as dental plaque (Figure 1) [1]. Dental plaque microbiota is highly complex and distinct according to the PD stage [4]. These bacteria can infiltrate the subgingival space and produce several metabolites such as ammonia and volatile sulfur compounds, which lead to halitosis as well as bacterial endotoxins and proteolytic enzymes, which promote periodontal inflammation and activation of the animal’s immune system [1]. The persistent host inflammatory response against the dental plaque bacterial aggression leads to PD progression [5,6].

It is known that PD presents a multifactorial aetiology, with genetic, microbiological, nutritional, and environmental factors displaying a high impact on its establishment. Additionally, PD incidence increases with age and in small breeds [5,7,8,9]. Considering PD prevalence, numbers ranging from 44 to 63.6%, rising to 84% in dogs that are 3 or more years old, or 100% in poodles that are more than 4 years old, have been described by Marshall et al. [8]. Recently, Stella and collaborators [9] determined a prevalence of 86.3%; similar values have been described in other reports [10,11,12]. The high prevalence of this disease in the canine population and its association with several severe complications, highlight the relevance of PD in veterinary medicine.

Considering disease progression, both local and systemic consequences can be linked to PD. Locally, PD can evolve to four stages, characterized by an increased periodontal destruction. Gingivitis, periodontal pocket formation, attachment loss with bone destruction, increased dental mobility, and furcation appearance are some of the local consequences of PD [11]. Systemically, bacteria and their toxins can spread via the bloodstream (bacteremia) from dental plaque, together with inflammatory mediators produced against periodontal bacterial aggression, leading to renal, hepatic, or cardiac consequences [5,13,14,15]. All of these can be dramatic to the animals’ health, rendering PD treatment and prevention essential.

Bacterial dental plaque control is the main key to manage this disease. The removal of dental plaque and the inhibition of its formation can be achieved by a combination of dental hygiene homecare procedures including toothbrushing and the application of several oral products, specific diet, the use of chew toys and regular professional periodontal procedures [2,3,16,17,18]. Associated with these procedures, host modulation therapy and antimicrobial therapy may be needed [11]. In some cases, periodontal surgery is essential to stop PD progression, minimize attachment loss, eliminate periodontal pockets, and promote periodontal regeneration [19].

## 2. Mechanical Plaque Reduction—Toothbrushing and Dental Scaling

Dental hygiene homecare can be performed using mechanical and chemical plaque reduction techniques. Toothbrushing is the most effective method for daily plaque control through the mechanical disruption of the dental plaque, being considered the gold standard method for PD control [12]. This procedure should be performed prior to PD establishment. Animals must be trained to accept brushing after the eruption of the permanent dentition, using positive reinforcement [11]. Some reports recommend a brushing technique with a soft-bristled nylon toothbrush, applying circular movements with the toothbrush held at a 45–60 angle to the tooth and used in a coronally directed stroke [1,20]. Ultrasonic toothbrushes may be used if the dog is trained to accept them.

The frequency of toothbrushing in dogs will influence the effectiveness in retarding dental plaque and calculus accumulation, and consequently, in reducing the severity of pre-existing gingivitis. According to Harvey (2015), brushing daily or every other day produces statistically significant improved results in dogs when compared with brushing weekly or every other week [21]. Therefore, it is recommended that this procedure is performed daily [12,22]. Additionally, in a 6 month randomized clinical trial, Allan and collaborators (2019) observed that daily toothbrushing is three times more efficient in dental plaque control when compared to the use of dental chews or specific dental diets, reinforcing toothbrushing as an optimal method for PD prevention [23].

Some dentifrices have incorporated chemical reduction compounds and attractive flavors in their composition to facilitate the removal of dental plaque and to improve animal toothbrushing acceptance [17,20,24]. In fact, the taste is part of the positive reinforcement that is important for establishing easier brushing behaviors. However, it is important to notice that, along with regular brushing procedures, professional periodontal evaluation and cleaning, that we will discuss later, are essential to prevent PD establishment. 

Dental scaling is a professional periodontal procedure aiming to remove the supragingival or subgingival dental plaque and calculus [24]. This process may be performed using a combination of manual scaling with scalers and curettes, and mechanical scaling with ultrasonic instrumentation. Allied to scaling, root planning, which is the removal of residual calculus from the root surface, polishing and sulcular lavage may be performed to reduce the periodontal microorganisms [11,18,24].

## 3. Chemical Plaque Reduction Products

As an adjunctive measure to the gold standard prophylactic method of toothbrushing, chemical plaque control agents are highly diverse and useful for PD control. These compounds interfere with biofilm composition and metabolism by disrupting the polymicrobial biofilm or by preventing its formation [25]. A major requirement for these agents is their capacity of retentiveness for maximum benefit. Several compounds are described as antiplaque chemical agents with diverse application methodologies, mechanisms of action, and effectiveness.

Chlorhexidine (CHX) is a cationic biguanide that disrupts the bacterial cell wall and precipitates the bacterial cytoplasm. It also has antifungal and antiviral properties [20]. CHX is used in solution to irrigate the oral cavity before dental scaling or surgical procedures, showing good activity against oral pathogens [22]. It is described as the substance with the greatest efficacy in the inhibition of oral plaque, being available in the form of solutions, gels, and toothpastes for home dental care [25,26]. Usually, it is used in solutions of 0.1 to 0.12% of chlorhexidine gluconate or acetate [11]. However, CHX presents negative effects when used as a prolonged therapy or at high concentration such as loss of taste, increased mineralization of the plaque, pigmentation of the enamel, or lesions of the oral mucosa [20,26]. In addition, chlorhexidine resistance and cross resistance with several antimicrobials are being documented in oral bacteria [27]. 

Zinc containing products are also interesting compounds in the fight against PD. Zinc compounds have antimicrobial activity against oral pathogens, inhibit calculus formation and reduce halitosis by binding to volatile sulfur compounds [11,28]. Frequently, zinc is combined with vitamin C (ascorbic acid), improving its activity by supporting collagen synthesis. However, it is described that a dietary zinc overload may occur from drinking zinc gluconate-containing products over long periods, which is linked to leucopenia, sideroblastic anemia, and digestive disorders [29].

Xylitol-based drinking water additives may also be used for dental plaque reduction [20]. Xylitol is a sugar alcohol compound used frequently in human dental products because of its anticaries effects [11,30]. Some toxic effects have been described in dogs, so caution should be taken in the administration of xylitol-based products [11,30,31]. On the other hand, erythritol, a sugar alcohol similar to xylitol, has been proposed as a promising compound to be used in canine PD prevention [30,32]. This compound shows the ability to inhibit pathogenic oral bacteria isolated from dogs with PD, being safe to be used in dogs [30,31,32].

Triclosan is a bisphenol with both antimicrobial and anti-inflammatory activities [33]. Its proposed mechanism of action includes bacterial membrane disruption and the inhibition of the oxygenase/lipoxygenase pathway [25,34]. Triclosan has the capacity to reduce dental plaque and gingivitis in humans, being safe and non-toxic [33]. In dogs, despite the potential use of this compound, there have been few studies focusing on the use of triclosan for PD control [35]. On the other hand, recent works have studied the antimicrobial ability of triclosan for the treatment of the surgical site and wound infections in dogs, and observed that this molecule can inhibit pathogens commonly isolated from wounds of dogs like *Staphylococcus pseudintermedius* including multidrug-resistant bacteria [36,37]. 

Nisin-biogel is a compound based on the antimicrobial activity of nisin and the delivery capacity of a guar gum biogel (Figure 2) [38]. Nisin is an antimicrobial peptide produced mostly by the bacterial species *Lactococcus lactis*, which presents in vitro antimicrobial activity against periodontal pathogens in the presence of saliva, both in humans and dogs [38,39,40,41]. Additionally, its safety regarding eukaryotic cells and reduced resistance development support the potential of this molecule for PD control [41,42]. Despite Howell et al. (1993) [43] showing that nisin could reduce canine gingivitis in a mouth rinse solution, and Cunha et al. (2021) [44] revealing that the nisin-biogel had slight effects on the canine commensal oral microbiome, new in vivo studies are essential to prove nisin-biogel efficacy.

Enzymes can be incorporated in toothpastes, mouth rinses, sprays, or gels to reduce dental plaque formation. The most frequent used molecules in veterinary medicine are glucose oxidase and lactoperoxidase. Glucose oxidase oxidizes glucose to gluconolactone and hydrogen peroxide, which activates the lactoperoxidase system. This system oxidizes thiocyanate to hypothiocyanite, which has antimicrobial activity [45]. Lysozyme and lactoferrin are also other enzymes frequently incorporated in toothpastes, snacks, or oral gels for dogs and cats. Lysozyme is a salivary antimicrobial compound that acts on the peptidoglycan of the bacterial cell wall, activates bacterial autolysins, and inhibits bacterial adherence [46]. Lactoferrin mainly presents antimicrobial activity by high affinity linkage with iron, making it unavailable to many bacterial species [47]. Other enzymes with potential and already used in human dentistry are polysaccharide hydrolases such as mutanases and dextranases that act on carbohydrate components of the biofilm matrix, and proteases, which disrupt bacterial adhesion to oral surfaces and affect cell-to-cell interactions [46]. However, the use of enzymes in oral health has some limitations including their sensitivity to proteolysis, low stability in oral hygiene products, and insufficient time of retention in the oral cavity [46].

Another chemical product described as effective in reducing calculus, dental plaque, halitosis, and gingivitis is cetylpyridinium chloride [25,48]. This compound is a cationic quaternary ammonium with broad spectrum activity that acts by disruption of the bacterial membrane [25]. There have been few studies investigating this molecule in dogs. However, Kim and collaborators (2008) showed that cetylpyridinium chloride was able to promote a reduction in dental plaque accumulation, calculus formation, and halitosis in dogs [48]. 

Fluoride is generally used in human dentistry because of its anticaries action [49]. It also has antibacterial activity, which is the main reason for its potential use in veterinary medicine. However, fluoride compounds have high toxicity, leading to a low applicability in animals, being used topically in specific cases [11].

## 4. Natural Products

Several studies have reported that natural products such as essential oils, herbal products, or algae may be beneficial for controlling PD, calculus, and halitosis [50,51,52,53,54,55,56].

Derived herbal products have been extensively explored in odontology. These compounds are usually included in toothpastes or mouthwash. Several reports have showed that distinct alcoholic herbal extracts present in vitro antimicrobial activity against oral pathogens, namely, extracts from *Eugenia uniflora*, *Rosmarinus officinalis*, *Myrciaria cauliflora*, *Nasturtium officinale*, *Lawsonia inermis*, *Malva sylvestris*, *Boswellia serrata*, *Salvia officinallis*, *Myristica fragrans*, *Myracrodruon urundeuva*, *Punica granatum*, and from the edible mushrooms *Lentinula edodes* or *Cichorium intybus* [53,57].

In addition, essential oils obtained from plants have also been reported to have in vitro antimicrobial activity against periodontopathogens such as oils obtained from *Lippia sidoides*, *Syzygium aromaticum*, *Cinnamomum zeylanicum, Cymbopogon citratus*, *Aloe vera* gel (Figure 3), *Matricaria recutita*, *Rosmarinus officinalis*, *Copaifera officinalis*, *Ferula assa*, *Rhododendron groenlandicum*, *Mentha piperita* L., or *Satureja montana* L. [50,51,53,55,56]. Beyond its in vitro activity, a mouth rinse with *Lippia sidoides* essential oil has been proven to reduce gingivitis and dental plaque index in dogs [50,51]. In addition, mouthwashes and toothpastes containing *Calendula officinalis* have been shown to present in vitro and in vivo antimicrobial activity against oral pathogens, with significant activity on gingival tissues [53]. *Schinus terebinthifolius* presented inhibitory activity in oral microorganisms and anti-inflammatory ability, revealing potential to be used in gingivitis treatment or control. In addition, the plant *Azadirachta indica* showed a significant reduction in gingival bleeding and dental plaque indices when tested as a mouthwash in humans [58]. 

Another interesting product is pomegranate (*Punica granatum*) extract, which has already shown antibacterial activity against oral bacterial strains from humans and dogs [59]. Its antimicrobial activity and antioxidant properties, mostly associated with polyphenols and tannins, suggest that pomegranate may be a useful compound to be used in canine PD control, being present in some water additives and dental chew compounds for dogs [59].

Propolis is also included within natural products with the potential to be used in PD control. Generally known as bee glue, propolis is a resinous substance accumulated by bees from different types of plants, mainly composed of waxes, polyphenols (phenolic acids, flavonoids), and terpenoids [60,61]. Studies have described its inhibitory activity toward oral pathogens as well as an effective activity in reducing gingivitis and halitosis [60].

Anther potential compound in PD control is chitosan, a copolymer obtained from chitin through enzymatic or chemical processes [62]. This molecule showed in vitro antimicrobial activity against several oral pathogens and the ability to reduce dental plaque when included in toothpastes or mouthwashes [53,62]. 

Pieri and collaborators (2016) [54] also showed that β-caryophyllene, a sesquiterpene obtained from plants, has antimicrobial activity against the dental plaque bacteria of dogs, and Laverty et al. (2015) [63] described that an emulsion of fatty acids also presented in vitro antimicrobial activity against resistant periodontopathogen biofilms implicated in canine and feline dental disease.

Finally, the brown alga, *Ascophyllum nodosum* has been described as effective in reducing dental plaque, calculus, gingival bleeding, and oral volatile sulfur compounds in dogs as well as in improving oral health, being commercially available as a powder supplement or dental treat for dogs [64,65].

## 5. Specific Dental Diet

Diet is one important factor in PD prevention. In fact, food texture and its nutritional composition may affect the oral environment, influencing oral tissue integrity, dental plaque establishment, and stimulation of the salivary flow [66]. Dry food promotes a mechanical abrasion in the dental surface, reducing dental plaque accumulation and calculus formation, while soft food such as the home diet is frequently associated with higher PD incidence [2,67,68]. Furthermore, an adequate nutritional composition is essential to maintain the oral tissue integrity. Logan (2006) stated that there are key nutritional factors that promote dental health in dogs [66]. First, protein deficiency may lead to periodontal degeneration, being indicated as a protein level of 16 to 35% for oral health maintenance. Then, minerals such as calcium and phosphorus are important nutritional components. Excess of these minerals may lead to calculus increase or even other systemic consequences. On the other hand, calcium deficiency and phosphorus excess has been related to nutritional secondary hyperparathyroidism, which may contribute to bone loss. The recommended food levels of calcium and phosphorus to promote oral health are 0.5–1.5% and 0.4–1.3%, respectively [66]. In addition, an increased kibble size and digestibility higher than 80% are also important [66].

Nowadays, a high number of commercial diets aiming at preventing dental plaque are available. Some of these diets include molecules that have some proven effects on PD control. For example, polyphosphates are usually a component of dental diets because of their activity as mineral chelators, preventing plaque mineralization and thus reducing the incidence of dental calculus [2,20,24].

In the fighting against PD, some diets may include components that promote a reduction in halitosis. For example, Di Cerbo and collaborators (2015) have shown that a diet composed by fish hydrolyzed proteins, *Ribes nigrum* L., *Salvia officinalis*, *Thymus vulgaris*, lysozyme, propolis, bioflavonoids, and vitamin C can promote a significant decrease in halitosis-related sulfur compounds in dogs [69]. 

The inclusion of vitamin C in diets may be beneficial for PD control. This nutrient can act as an antioxidant, as a cofactor for enzymes, in capillary perfusion, and by inducing the differentiation of periodontal ligament cells [70,71]. Human studies suggest that vitamin C contributes to a reduced risk of periodontal disease [70].

## 6. Chew Toys and Dental Treats

In addition to specific food, chew toys, biscuits, bovine skin treats, and bones can be included in the daily home dental care for promoting oral health (Figure 4) [72]. These materials act by stimulating salivation and teeth abrasion, leading to the mechanical removal of the dental plaque [20]. Some of them may include chemical anti plaque agents in their composition to enhance their efficacy. Quest (2013) showed that adding a dental chew to the regular diet of dogs for 28 days resulted in statistically significant reductions in dental plaque and calculus accumulation as well as improved halitosis and gingival indices [73]. The same result was obtained by Mateo et al. (2020), Carroll et al. (2020), and Oba et al. (2021), who evaluated the effect of different dental chews on the oral health of dogs [74,75,76]. In addition, Stookey (2009) showed that the daily ingestion of a soft rawhide chew containing sodium tripolyphosphate and cetylpyridinium chloride resulted in a 28% reduction in calculus formation, 18.5% reduction in dental plaque, and 45.7% reduction in gingivitis [77]. Pinto et al. (2020) analyzed the impact of two autoclaved bovine bones with distinct hardness, a cortical raw compact and a spongy bone, on oral and dental health as well as in dental calculus reduction in adult dogs [78]. Both bone types used in their study were highly effective in removing dental calculus, with a higher reduction in spongy bone (86%) [78]. 

Recently, Gawor and collaborators (2021) performed a comparison of different chews regarding the maintenance of good oral health in dogs and observed that all chews tested provided a reduction in dental plaque accumulation [72]. However, the vegetable-based dental chew tested revealed better results, which emphasizes the importance of selecting the best products according to the existing information [72]. 

However, chewing products may have problems, as excessive abrasion may promote the migration of the gingival margin in the apical direction, leading to gingival fissures; and inappropriate chewing of hard chew toys such as bones, antlers, hard nylon bones, cow hooves, and other devices may cause tooth fractures, resulting in endodontic disease or even in oral and esophageal lesions and intestinal obstructions [20,78]. The selection and use of chew devices should be performed according to the age, size, and health status of the dog, avoiding hard materials. It is important to highlight that chew toys or other dental treats do not replace daily toothbrushing, which is the gold standard procedure for periodontal disease prevention. These items are always additional measures that can be used to promote dental health [11,78].

## 7. Host Modulation Therapy

The immunity system has an important role in PD establishment. As referred, an inflammatory response to the periodontal aggression promoted by the microorganisms in the dental plaque is one of the keystones of PD [1,2,11]. This response includes the release of pro-inflammatory cytokines, prostaglandins, or enzymes like metalloproteinases, which induce catabolic processes such as bone resorption and collagen destruction, resulting in periodontium destruction [3,5,79]. A stimulation of neutrophil chemotactic attraction, the activation of the complement, arachidonic acid and kinin cascades, and the induction of mast cell degranulation also occur [1,2,3]. Bearing these aspects in mind, anti-inflammatory drugs that act on these processes may be beneficial for PD management. Some reports have described that the systemic use of non-steroidal anti-inflammatory drugs (NSAIDS) may be included in the treatment of severe PD cases, but always together with professional periodontal procedures and dental hygiene homecare. The daily use of NSAIDS can inhibit prostaglandin synthesis and slow the rate of alveolar bone loss; however, long treatment periods with these agents may lead to several side effects [11]. On the other hand, the topical application of antimicrobials such as doxycycline and azithromycin at sub inhibitory concentrations have shown anti-inflammatory activity in PD cases [11,80]. In humans, systemic sub antimicrobial concentrations of tetracyclines, especially doxycycline, are approved for use in the management of several diseases including PD [81]. The use of these compounds relies on their powerful activity as matrix metalloproteinase inhibitors, interfering with the collagenolytic activity of metalloproteinases and reducing PD progression [81]. Despite the use of subinhibitory doxycycline concentrations, several studies have concluded that its application in long- or short-term regimens showed no impact on the development of antimicrobial resistant bacteria [81]. However, a second generation of host modulation agents, based on the chemical modification of tetracyclines focusing on the removal of their antimicrobial activity and preservation of their metalloproteinase inhibition capacity, has been studied. These compounds have been submitted to clinical trials, revealing modest efficacy and some adverse effects, showing no major benefits for PD control [81].

Additionally, compounds that help to contain or control the excessive local inflammatory reaction by reducing neutrophil recruitment and blocking pro-inflammatory cytokines or reactive oxygen products may be useful, and include endogenous lipoxins, lipid-inflammatory mediators, pentoxifylline, cimetidine, or mercaptoethylguanidine [11].

Other host modulation agents that are being studied for PD treatment are chemical modified curcumins. These molecules also act as inhibitors of matrix metalloproteinases, having proven beneficial effects on PD in in vivo studies by preventing alveolar bone loss and reducing proinflammatory cytokines and matrix metalloproteinases in the gingival tissues [81].

There are several compounds in study with the potential to act as host modulating agents for PD treatment, such as resveratrol, omega-3 polyunsaturated fatty acids, eicosapentaenoic acid, docosahexaenoic acid, antibodies against the pro-inflammatory cytokines, and cimetidine [81,82].

To assure beneficial results in PD control, all of these host modulation compounds must be used in combination with non-surgical periodontal therapy, such as scaling and root planning [11].

## 8. Antimicrobial Therapy

The use of systemic antibiotics for PD treatment should be evaluated on a case-by-case basis; however, it is usually not recommended in cases of gingivitis and mild periodontitis [11,22]. In severe cases of PD and/or patients with systemic risk factors, systemic antimicrobial administration may be used to reduce the oral bacterial load and the risk of bacteremia [22]. Some antimicrobial drugs administered to dogs with PD are amoxicillin/clavulanate, clindamycin, and metronidazole [3,22].

On the other hand, topical antimicrobials may be indicated in severe cases of periodontitis, with periodontal pockets deeper than 5 mm [22]. Topical gels, also known as periceutic gels, containing clindamycin or doxycycline, are some examples of compounds that can also be used [3,11,83].

## 9. Other Management Options

Antimicrobial photodynamic therapy is a laser-based technique, widely used in human dentistry, as an adjuvant approach for the treatment of periodontal or endodontic infections [84]. This therapy produces a bactericidal effect by producing reactive oxygen species, has an antibiofilm activity against oral mature biofilms, and contributes to the healing of periodontal tissues [84,85]. In dogs, antimicrobial photodynamic therapy was used in ligature-induced peri-implantitis, and successfully decontaminated the infected surfaces [86]. This is a promising technique that can be used by trained veterinarians as an adjunctive treatment in dental procedures.

Another interesting approach in canine PD control is probiotics. Some authors have stated that probiotics may act on the maintenance of a healthy oral microbiota, helping in the prevention of periodontitis, and can be used in combination with conventional treatments in established PD cases [87,88]. You and collaborators (2022) characterized several *Lactobacillus acidophilus* strains, and found nine with antimicrobial activity against caries-causing pathogens [88]. One of them was considered specific and safe for dogs, with the potential to be applied as a specific probiotic strain for oral health in dogs [88].

Other important products for PD management are dental sealants. These products are applied to the tooth surface, providing protection of the dentin tubules, which leads to an internal healing of the teeth [89]. In veterinary medicine, sealants are also used to reduce dental plaque accumulation [22]. These barrier dental sealants are waxy polymers applied first by professionals (sealants with high viscosity). In addition, there are sealants with low viscosity that can be applied by the pet owner weekly in some specific cases [20,24]. The application of resin-bonded sealants, usually used for fractured teeth, must be carried out by a trained veterinarian using intraoral radiographs to evaluate tooth vitality over time [22].

In 2007, a periodontal vaccine targeting *Porphyromonas* was available for administration in dogs for PD control [20,24]. However, in 2011, its commercialization was cancelled because vaccinated dogs showed no reduction in PD progression in comparison with unvaccinated dogs [18].

## 10. Professional Periodontal Procedures and Treatments

A regular oral evaluation is essential for preventing PD in dogs. At the first veterinary visit, after the examination of the animal, it is important to explain to the owner the relevance of oral health, including the individualized prevention strategies that should be maintained on an ongoing basis [22]. Home oral hygiene training can be started in dogs after eruption of the permanent dentition. According to the dental care guidelines for dogs and cats published by the American Animal Hospital Association, it is recommended to perform a complete dental prophylaxis (complete dental cleaning, polishing, and intraoral dental radiographs) by 1 year of age for small and medium-breed dogs, and by 2 years of age for larger-breed dogs. Then, the animal’s dental health should be evaluated every 6 to 12 months [22].

This regular professional periodontal evaluation includes a complete historical and physical examination including an assessment of all body systems and a conscious oral evaluation of the animal. Next, a complete oral evaluation with the anesthetized animal is mandatory, with tooth-by-tooth visual examination, probing, mobility assessment, radiographic examination, and oral exam charting [22]. This evaluation will allow for PD detection and classification, in order to decide on the adequate treatment.

Professional periodontal treatments may include surgical or non-surgical procedures. The first line of treatment is always non-surgical and includes the removal of factors that promote disease progression, dental scaling, root planning, and polishing [24]. Surgical procedures such as gingivectomy, gingivoplasty, open flap debridement, or osseous grafting, may be used to gain access to the root surfaces and to remove local factors that are promoting PD development [2,18,24,90]. Guided periodontal regeneration (GPR) techniques can be used in cases with periodontal tissue destruction. In GPR, a barrier membrane is used to prevent apical migration of the junctional epithelium, promoting the regeneration of the tooth supporting tissues [2,90,91]. There are several in vitro and in vivo studies with a focus on new compounds to be used in severe cases with bone destruction as GPR including collagen barrier drug delivery membranes, lovastatin–chitosan–tetracycline nanoparticles, fibroblast growth factor with collagen hydrogel, bone marrow stem cells with collagen–hydroxyapatite, mesenchymal stem cells, hyaluronic acid in a collagen matrix, and low intensity pulsed ultrasound [92,93,94,95,96].

When periodontal regeneration techniques and associated dental hygiene care cannot be performed or show no results, the final procedure is tooth extraction [3,24]. 

## 11. Conclusions

A high prevalence of PD in dogs and its relation with potential local and systemic consequences reinforce the need for investment in new preventive measures, treatments, and oral procedures to improve the control of this disease in these animals. There are several available products that can be used to prevent PD in dogs, a regular professional periodontal evaluation being essential to define the best strategy to apply to each animal. Scientific research is being performed in order to develop new compounds or to repurpose products to be used in PD control.

Knowledge of the specific guidelines and variety of available products and procedures are fundamental to apply the most accurate treatment to each dog with PD.

## Figures and Tables

**Figure 1 antibiotics-11-01729-f001:**
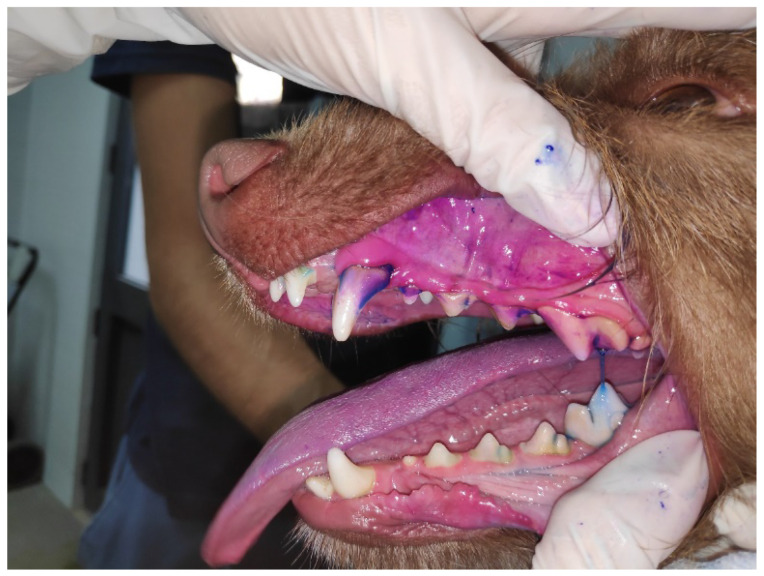
Observation of the dental plaque in the oral cavity of a dog, by using a disclosure solution.

**Figure 2 antibiotics-11-01729-f002:**
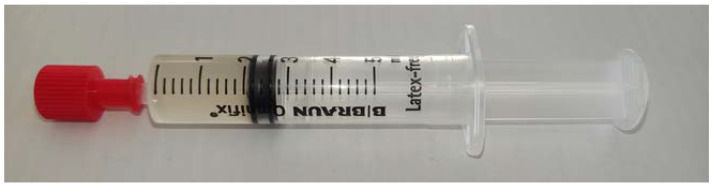
Single dose of nisin-biogel.

**Figure 3 antibiotics-11-01729-f003:**
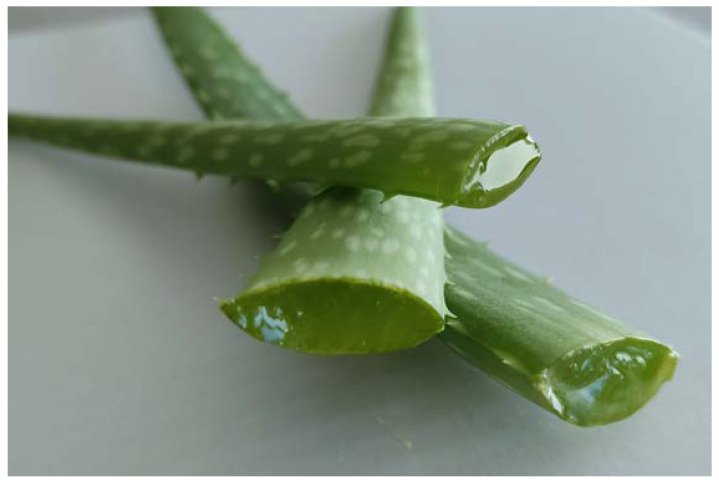
*Aloe vera*.

**Figure 4 antibiotics-11-01729-f004:**
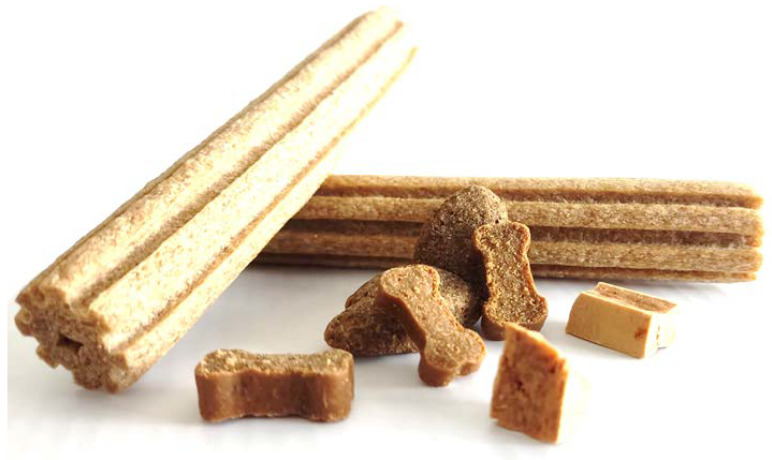
Dental treats and biscuits for dogs.

## Data Availability

Not applicable.

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
