# Peer review of "Revisiting Periodontal Disease in Dogs: How to Manage This New Old Problem?"

_antibiotics, 2022, doi:10.3390/antibiotics11121729_

Round 1

Reviewer 1 Report

This study reviews the PD management and prevention methods for dogs. There are several issues that authors need to address before the manuscript could get accepted.

Third paragraph in “1. Introduction”, line 5, PD can lead to “oronasal fistulas, ocular disturbances”, but “mandibular fractures” are very rare. Please remove this outcome.

Fourth paragraph in “1. Introduction”, line 4, barrier dental sealants are used for caries prevention, not PD prevention.

The second section head is recommended to be changed to “2. Mechanical plaque reduction-toothbrushing and dental scaling”. Or, 2.1 as “homecare-toothbrushing” and 2.2 as “professional treatment-dental scaling and root planning”. Please also add the related contents of dental scaling to the manuscript.

Fifth paragraph in “3. Chemical plaque reduction products”, last sentence, what does “interesting (result)” mean? Please specify.

Does “4. Natural products” belong to “chemical products”? Please reorganize the heading/subheadings according to the contents.

Second paragraph in “9. Antimicrobial therapy”, the periodontal pockets depth more than 5 mm is severe periodontitis, not mild/moderate periodontitis.

Author Response

Dear Reviewer,

We would like to thank you for the comments and the discussion points presented. We agree that they will improve the quality and understanding of the manuscript.

As suggested, we removed the sentences about dental sealants and “oronasal fistulas…”  in the introduction section and performed several changes all over the manuscript that are highlighted in yellow.

The second section was changed to “2. Mechanical plaque reduction-toothbrushing and dental scaling”, and the text was changed according to the reviewer’s suggestion.

Considering the topic “natural products”, we believe that this topic should remain in a different section because it includes information regarding herbal products, oils or algae, which does not fit under the term chemical products.

Best Regards,

Eva Cunha

Reviewer 2 Report

This review is interesting , well-organized and well-written. This reviewer suggests authors add some figures to improve readability. 

Author Response

Dear reviewer,

We would like to thank you for your comment. As suggested, we added four figures to improve the readability of the document.

Best Regards,

Eva Cunha

Reviewer 3 Report

The manuscript entitled "Revisiting periodontal disease in dogs: how to manage this 2 new old problem?" is a very interesting review of periodontal disease. Although well presented and written, does not bring anything new to the field except that can be an easy way for the clinician to consult this information. I am not sure if this paper is suitable for this journal, although periodontal diseases need antibiotics for their treatment, the reference to this treatment is only a small portion of the paper and nothing new is added.

Author Response

Dear reviewer,

Thank you for your comment.

This manuscript corresponds to a review about the procedures available to control PD in dogs. In these animals, PD control is focused on prevention, and there are several strategies and products that can be used with that purpose, which act mostly as antimicrobials. So, despite the fact that the treatment of this disease includes antimicrobials and removal of the dental plaque, the main objective is to control the disease, by acting on dental plaque establishment. There are many non-conventional antimicrobial compounds that allow to achieve that, that we describe over this review. For that reason, we believe that this could be an interesting manuscript to be published in the Special Issue “Antimicrobial Agents in Oral Diseases: Prophylaxis and Therapy between New and Old Molecules” of the journal “Antibiotics”, since it aims to publish papers on the prophylaxis in oral diseases. Also, in our opinion, this review will be a valuable instrument to be used by clinicians or academic professionals in their daily work.

Best Regards,

Eva Cunha

Round 2

Reviewer 3 Report

To the manuscript Revisiting periodontal disease in dogs: how to manage this 2 new old problem? under special Issue “Antimicrobial Agents in Oral Diseases: Prophylaxis and Therapy between New and Old Molecules” of the journal “Antibiotics” can fit the aims and scopus of the journal.